# Long-range Brain Graph Transformer

**Shuo Yu**[1,2], **Shan Jin**[3], **Ming Li**[4,5], **Tabinda Sarwar**[6], **Feng Xia**[6✉]

[1]School of Computer Science and Technology, Dalian University of Technology, China
[2]Key Laboratory of Social Computing and Cognitive Intelligence (Dalian University of Technology),
Ministry of Education, China
[3]School of Software, Dalian University of Technology, China
[4]Zhejiang Institute of Optoelectronics, China
[5]Zhejiang Key Laboratory of Intelligent Education Technology and Application,
Zhejiang Normal University, China
[6]School of Computing Technologies, RMIT University, Australia
`{shuo.yu, f.xia}@ieee.org`
`jinshan0924@mail.dlut.edu.cn`
`mingli@zjnu.edu.cn`
`tabinda.sarwar@rmit.edu.au`

## Abstract

Understanding communication and information processing among brain regions
of interest (ROIs) is highly dependent on long-range connectivity, which plays a
crucial role in facilitating diverse functional neural integration across the entire
brain. However, previous studies generally focused on the short-range dependen-
cies within brain networks while neglecting the long-range dependencies, limiting
an integrated understanding of brain-wide communication. To address this limi-
tation, we propose **A**daptive **L**ong-range aware **T**ransform**ER** (ALTER), a brain
graph transformer to capture long-range dependencies between brain ROIs utilizing
biased random walk. Specifically, we present a novel long-range aware strategy
to explicitly capture long-range dependencies between brain ROIs. By guiding
the walker towards the next hop with higher correlation value, our strategy simu-
lates the real-world brain-wide communication. Furthermore, by employing the
transformer framework, ALERT adaptively integrates both short- and long-range
dependencies between brain ROIs, enabling an integrated understanding of multi-
level communication across the entire brain. Extensive experiments on ABIDE
and ADNI datasets demonstrate that ALTER consistently outperforms generalized
state-of-the-art graph learning methods (including SAN, Graphormer, GraphTrans,
and LRGNN) and other graph learning based brain network analysis methods
(including FBNETGEN, BrainNetGNN, BrainGNN, and BrainNETTF) in neuro-
logical disease diagnosis. Cases of long-range dependencies are also presented to
further illustrate the effectiveness of ALTER. The implementation is available at
`https://github.com/yushuowiki/ALTER`.

## 1   Introduction

Brain networks represent a blueprint of communication and information processing across different
regions of interest (ROIs) [1, 2]. The interaction between anatomically connected ROIs within brain
networks is the foundation of brain network analysis tasks [3]. As shown in Figure 1, numerous
studies have shown that brain networks exhibit not only short-range connectivity (i.e., short-range
dependencies) but also extensive long-range connectivity (i.e., long-range dependencies) [4, 5, 6, 7].
Short-range dependencies rely on the neighbourhood space, whereas long-range dependencies

38th Conference on Neural Information Processing Systems (NeurIPS 2024).

reflect long distance communication among ROIs. Such long-range dependencies play a vital role in theoretical analyses of brain function [8], dysfunction [9], organization [10], dynamics [5], and evolution [11]. Therefore, it is necessary to capture long-range dependencies within brain networks to better represent communication connectivity and facilitate brain network analysis tasks to extract valuable insights. The communication connectivity is also known as functional connectivity, representing the interaction between brain ROIs.

Several existing studies have been devoted to representing communication connectivity within brain networks via graph learning methods for network analysis tasks [12, 13, 14, 15]. However, as previously mentioned, they generally focus on the aggregation of neighborhood information, i.e., short-range dependencies, which still limits their effectiveness by neglecting the crucial long-range dependencies. Most of the studies build models to analyze node (ROIs) features and structures (inter-ROI connectivity) using Graph Neural Networks (GNNs) with message-passing mechanism. The limited expressiveness of GNNs fails to capture the long-range dependencies in brain networks. While the group-based graph pooling operations cluster ROIs, these are still limited to regional similarities and are not sufficient to represent long-range communication connectivities.

To address this limitation of solely considering the short-range dependencies, we aim to develop a solution that leverages long-range dependencies to enhance brain network analysis tasks. Currently, the capturing the long-range dependencies has been addressed in different network analysis tasks outside the scope of brain studies, such as those related to social networks and molecular networks [16, 17, 18, 19]. Among these, random walk methods are widely adopted, as they can explicitly capture long-range dependencies by aggregating structure information across the entire

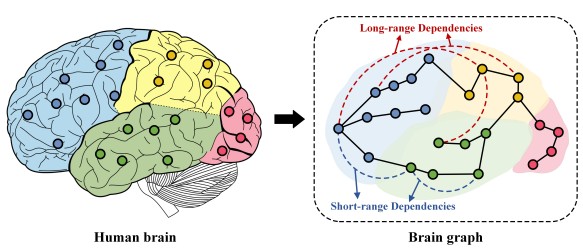

Figure 1: An illustration of long-range dependencies and short-range dependencies within human brain.

random walk sequence [20, 21, 22, 23]. In conventional random walk methods, the transition probability from a node to one of its neighbors is typically uniform. However, several studies have observed that different pairs of ROIs typically demonstrate varying communication strengths in brain activity, where stronger communication indicates greater dependencies among ROIs [24, 25, 26]. Therefore, employing conventional random walk methods to sample the next hop with uniform probability renders it impossible to capture the long-range dependencies within brain networks.

Based on the above observation, we are dedicated to capturing long-range dependencies in brain networks under the varying communication strengths among ROIs. In this paper, we propose **A**daptive **L**ong-range aware **T**ransform**ER** (ALTER), a brain graph transformer to capture long-range dependencies between brain ROIs extracted from biased random walk. Specifically, in order to capture long-range dependencies within brain networks, we firstly present an **A**daptive **L**ong-ran**G**e **A**ware (ALGA) strategy based on random walk in Section 3.1, which explicitly samples random walk sequences based on varying communication strengths among ROIs. In this strategy, we initially calculate the inter-ROI correlations as adaptive factors to evaluate their communication strengths. Subsequently, the use of random walk is biased, subject to the next hop with a higher correlation value, thus explicitly encoding long-range dependencies as long-range embeddings through random walk sampling. Furthermore, given the significance of both short-range and long-range dependencies in brain network analysis tasks, we introduce an effective brain graph transformer in Section 3.2, which can capture different levels of communication connectivities in human brains. Specifically, we inject the long-range embeddings into a transformer framework and integrate both short-range and long-range dependencies between ROIs using the self-attention mechanism.

The contributions of the paper are summarized as follows: 1) pioneering the explicit emphasis on the significance and challenges of capturing long-range dependencies in brain network analysis tasks, we propose a novel solution for capturing long-range dependencies within brain networks; 2) to address the limitations of previous studies that overlook long-range dependencies within brain networks, we introduce a novel brain graph transformer with adaptive long-range awareness, which leverages the communication strengths between ROIs to guide the capturing of long-range dependencies, enabling an integrated understanding of multi-level communication across the entire brain; 3) extensive

experiments on ABIDE and ADNI datasets demonstrate that ALTER consistently outperforms generalized graph learning methods and other graph learning-based brain network analysis methods.

## 2 Related Work

### 2.1 Brain Network Analysis

Several studies have developed graph learning-based methods for brain network analysis tasks, such as neurological disease diagnosis and biological sex prediction [12, 14, 15, 27, 28]. The majority of studies have utilized GNNs to learn the information of ROIs and inter-ROI connectivity. For instance, Li et al. [12] utilized GNNs with ROI-aware and ROI-selection to perform community detection while retaining critical nodes. Kan et al. [28] dynamically optimized a learnable brain network. Additionally, a few models based on specific graph pooling operations were also proposed to retain the communication information of brain networks. Specifically, Yan et al. [14] designed group-based graph pooling operations to enable explainable brain network analysis. Kan et al. [15] considered the similarity property among brain ROIs and designed a graph pooling function based on clustering. However, these approaches are generally limited to the aggregation of neighborhood information, while neglecting the long-range connectivity that plays a key role in brain network analysis tasks [5].

### 2.2 Graph Transformer

Several existing studies have focused on developing the transformer variants for graph representation learning [29, 30, 31, 32, 33]. Transformers have demonstrated competitive or even superior performance over GNNs. Dwivedi et al. [29] were the first to extend the transformer to graphs, defining the eigenvectors as positional embeddings. Kreuzer et al. [30] improved the positional embeddings and enhanced the transformer model by learning from the full Laplacian spectrum. Ying et al. [31] embedded the structural information of graph into a transformer, yielding effective results. Moreover, some studies have applied transformers to address unique issues in general graphs or domain-specific graphs. Wu et al. [32] utilized global attention to capture long-range dependencies within general graphs. Tao et al. [33] employed the transformer model to integrate both temporal and spatial information in social networks for disease detection.

## 3 Method

Within the brain graph, the collection of brain ROIs serves as the node set, and the features of these ROIs serve as the node features. The connectivity among brain ROIs is generally represented by the adjacency matrix. In this paper, we focus on analyzing these brain graphs for neurological disease diagnosis. Formally, consider a set of subjects' brain network $\{G_1 \ldots G_L\} \subseteq \mathcal{G}$ and their disease state labels $\{y_1 \ldots y_L\} \subseteq Y$, where $L$ is the total number of individuals (size of the dataset). Each brain graph $G$ contains $N$ ROIs, defined as $G = (V, X, A)$, where $V$ is node set, $X \in \mathbb{R}^{N \times d}$ are node features with dimension $d$, and $A \in \mathbb{R}^{N \times N}$ is an adjacency matrix. We aim to learn a representation vector $h_G$ that will allow us to predict the disease state of brain graph $G$, i.e., $y_G = f(h_G)$ where $f$ is prediction function. Notably, the proposed method can also deal with other brain network analysis tasks such as biological sex prediction.

The overall framework of ALTER is illustrated in Figure 2. Briefly, we first extract the node features $X_G$ and adjacency matrix $A_G$ from the fMRI data. Subsequently, adaptive factors $F_G$ are calculated using the temporal features of the fMRI data. Next, using the adaptive factors $F_G$ and the adjacency matrix $A_G$, the long-range embedding $E_G$ representing the long-range dependencies is obtained through adaptive long-range encoding. The encoding is utilized by our **A**daptive **L**ong-ran**G**e **A**ware (ALGA) strategy to explicitly encapsulate the long-range dependencies among ROIs as long-range embedding $E_G$. Finally, the long-range embedding $E_G$ is injected into the self-attention module, and a graph-level representation of the brain network is generated using the readout function to the downstream tasks. The complete training process is supervised by the cross-entropy loss.

Detailed description of these steps are discussed in the upcoming sections.

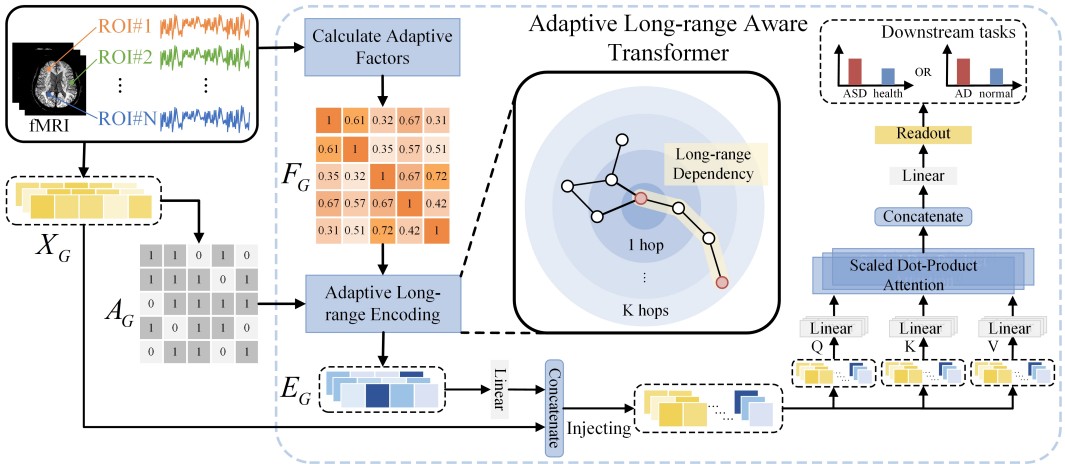

Figure 2: The overall framework of the proposed ALTER.

## 3.1 Adaptive Long-range Aware (ALGA) Strategy

As previously mentioned, previous studies primarily focus on aggregating information from neighboring ROIs, generally neglecting their long-range connectivity. In light of this, our goal is to design an efficient strategy to capture long-range dependencies among the ROIs. We now explain how the adaptive long-range aware strategy achieves this through the computation of adaptive factors and the adaptive long-range encoding.

### 3.1.1 Adaptive Factors

The correlation between ROIs reflects their communication strength within specific time frames, which is crucial for comprehending the functional organization, dysfunctions, and information propagation in the brain [2, 34]. Neuroscientific investigations have unveiled the existence of phase synchrony in neural oscillations among distinct ROIs, closely associated with the occurrence of perceptual motor behaviors and the integration of brain functional organization [24, 25, 26]. The degree of phase synchrony indicates the strength of connectivity between ROIs, whereby higher phase synchrony signifies stronger communication and consequently higher correlation values among the respective ROIs. This manner simulates real-world brain-wide communication.

Considering the above observation, we first calculate the correlation between ROIs as adaptive factors $F_G \in \mathbb{R}^{N \times N}$ to evaluate the communication strengths between ROIs. Specifically, the adaptive factor $f_{ij}$, denoting the communication strength between node $i$ and node $j$ within the brain graph $G$, is defined as:

$$f_{ij} = \begin{cases} \frac{\mathrm{Cov}(t_i, t_j)}{\sigma(t_i)\sigma(t_j)}, & \text{if } v_i \text{ and } v_j \text{ are connected,} \\ 1, & \text{if } i = j, \\ 0, & \text{otherwise,} \end{cases} \tag{1}$$

where $t_*$ denotes the raw feature of the brain ROI $v_*$ (e.g., temporal feature of fMRI data). $\mathrm{Cov}(\cdot)$ and $\sigma(\cdot)$ denote covariance and variance operations, respectively. The adaptive factors $F_G$ will influence the exploration mechanism of the random walk in the adaptive long-range encoding (Section 3.1.2). Specifically, ROIs with stronger connectivities exhibit higher transfer probabilities of the next hop compared to ROIs with weaker connectivities in the random walk. Note that for simplicity, we choose Pearson correlation coefficient to define the adaptive factors, without any modification.

### 3.1.2 Adaptive Long-range Encoding

The adaptive factors between ROIs, computed in the preceding step, are crucial for effectively capturing long-range dependencies between brain ROIs. Motivated by the random walk methods [16, 17, 35], we approach the problem of capturing long-range dependencies as a structural encoding task. By sampling and encoding node sequences into embeddings, we effectively capture long-range

communication between brain ROIs. Hence, we compute a long-range embedding using the walker sampling of node sequences under the constraint of adaptive factors.

In a network, a random walk involves transitioning from one node to another. Specifically, when a walker moves from node $i$, the probability of the walker moving to node $j$ in the next step depends solely on the conditions of nodes $i$ and $j$. This characteristic, where the probability of reaching node $j$ is independent of the preceding step at node $i$, defines a Markov process. Thus, the random walk process inherently embodies a Markovian nature.

Let $p_{ij}$ represent the probability that the walker walks from node $i$ to node $j$ in brain graph $G$, then $p_{ij}$ can be represented in the following matrix form:

$$P_G = \begin{bmatrix} p_{11} & \cdots & p_{1n} \\ \cdots & \cdots & \cdots \\ p_{n1} & \cdots & p_{nn} \end{bmatrix}, 0 \leq p_{ij} \leq 1, \sum_{v=1}^{n} p_{ij} = 1, \tag{2}$$

where matrix $P_G$ is the transfer matrix of brain graph $G$. Then the state vector is defined as:

$$T(k) = (t_1(k), t_2(k), \ldots, t_k(k)), \sum_{j=1}^{n} t_j(k) = 1, \tag{3}$$

where $k$ is denoted as the number of hops in random walks. $t_j(k)$ is the probability of the walker stops at node $j$ after $k$ times walk. $t_j(k)$ is called $k$ steps state probability. According to the total probability formula, we get:

$$t_i(k+1) = \sum_{j=1}^{n} t_j(k) p_{ij}, k = 0, 1, 2, \ldots, K. \tag{4}$$

where $K$ represents the total number of hops in random walk. So, we get the general recursive formula as $T(k) = T(0) P_G^k$. Nevertheless, brain networks deviate from general networks as pairwise ROIs typically exhibit distinct communication strengths, indicative of the collaborative nature of ROIs in brain activity. Falsely treating pairs of ROIs with varying communication strengths equally may disrupt the collaborative dynamics within brain activity. Given this fact, the transfer probability of the walker in a brain network can be fine-tuned using the adaptive factors, i.e., $\hat{P}_G = F_G \odot P_G$, where $\odot$ denotes the dot product operation. Formally, considering the adjacency matrix $A_G$ and the corresponding diagonal degree matrix $D_G$ of a brain graph $G$, along with the obtained adaptive factors $F_G$, we define the random walk kernel $R$ for adaptive long-range encoding as follows:

$$R = (F_G \odot A_G) D_G^{-1}. \tag{5}$$

In particular, The introduction of degree matrix can help to obtain richer information about brain-wide communication and is very commonly used in network analytics [36]. Since the degree matrix provides the number of degrees for each ROI, its ability to reflect the active state of the ROI in communication is important in determining which ROIs play a key role in information propagation. Hence, a degree matrix determines the transfer probability of a node to its neighboring nodes and highly influences the behavior of walker [37].

In the $K$-step random walk, the long-range embedding $E_G$ initialized by the adaptive long-range encoding is defined as:

$$e_i = \left[ I, R, R^2, \ldots, R^{K-1} \right]_{ii} \in \mathbb{R}^K, \tag{6}$$

where $I$ denotes the identity matrix. $e_i$ denotes a long-range embedding associated with the $i$-th node, encapsulating the long-range dependency asscoiated with $i$-th node. Through adaptive long-range encoding, we can explicitly capture long-range dependencies among ROIs in the brain graph $G$ and encode them into the form of long-range embeddings $E_G$.

## 3.2 Long-range Brain Graph Transformer

In Section 3.1.2, we obtained the long-range embedding $E_G$ that explicitly encode the long-range connectivities within the brain network $G$. As short-range dependencies are also significant, we aim to present an effective brain graph transformer by first injecting long-range embeddings into brain network representation learning and then integrate both long-range and short-range dependencies for learning a more comprehensive representation. To achieve this objective, we begin by describing the process of injecting long-range embeddings $E_G$ into the brain graph transformer. Later, we explain how the self-attention mechanism can be utilized to integrate long-range and short-range dependencies among brain ROIs.

**Injecting Long-range Embedding.** The computed long-range embeddings $E_G$ should be injected into the brain network transformer in a manner that enhances its utility. To acheive this, we introduce a fine-tuning procedure aimed at enhancing long-range embeddings $E_G$ and injecting them into the brain graph transformer. Specifically, we utilize a linear layer as a remapping function for long-range embeddings $E_G$, facilitating the injection of long-range dependencies within the brain network. This process enables the acquisition of trainable long-range embeddings $\hat{E}_G$ with dimension $k'$. Formally, this procedure is defined as:

$$\hat{E}_G = \text{LL}(E_G; W_G) = W_G E_G + b_G \in \mathbb{R}^{N \times k'}, \tag{7}$$

where $W_G \in \mathbb{R}^{k' \times k}$ and $b_G \in \mathbb{R}^{k'}$ denote learnable weight matrix and bias vector, respectively.

**Self-attention Module.** Transformer-based models generally surpass conventional representation learning methods in their ability to capture pairwise token correlations and the influence of individual tokens. This stems from the self-attention mechanism's capability to allow inter-token communication. Nonetheless, employing initial node features as input tokens is insufficient for Transformer-based models to effectively capture complex inter-dependencies within brain networks. Furthermore, pairwise ROIs often exhibit varying degrees of short- and long-range dependencies across various brain network analysis tasks [38, 39, 40, 41]. Hence, we need to integrate both long-range and short-range communication among ROIs through a self-attention module. To model this mechanism, we begin by constructing tokens through the combination of learnable long-range embeddings $\hat{E}_G$ and initial node features $X_G$. Then, we utilize a vanilla transformer encoder as the framework for the self-attention module.

Formally, we concatenate learnable long-range embeddings $\hat{E}_G$ and initial node features $X_G$ as tokens $\hat{X}_G$, and then utilize a transformer encoder with $L$-layer nonlinear mapping and $M$ attention head to learn comprehensive node features $Z_G$:

$$\hat{X}_G = \left[ X_G \,\middle|\, \hat{E}_G \right] \in \mathbb{R}^{N \times \left(d + k'\right)}, \tag{8}$$

$$Z_G = W_o \left( \|_{m=1}^M Z_G^{m,l} \right) \in \mathbb{R}^{N \times d_{out}}, \ Z_G^{m,l} = \text{softmax} \left( \frac{Q^{m,l} K^{m,l^T}}{\sqrt{d_{out}^{m,l}}} \right) V^{m,l} \in \mathbb{R}^{N \times d_{out}^{m,l}}, \tag{9}$$

with $Q^{m,l} = W_q Z_G^{m,l-1}$, $K^{m,l^T} = \left( W_k Z_G^{m,l-1} \right)^T$, and $V^{m,l} = W_v Z_G^{m,l-1}$ are the query matrix, the key matrix, and the value matrix, where $Z_G^0 = \hat{X}_G$, $\|$ and $[\cdot | \cdot]$ both indicate the concentrate operation, $l$ and $m$ denote the layer index and the head index, $W_q, W_k, W_v \in \mathbb{R}^{d_{out}^{m,l} \times d_{out}^{m,l-1}}$ and $W_o \in \mathbb{R}^{d_{out} \times d_{out}^m}$ are learnable projection matrices. In the representation learning procedure, the employed Transformer framework enables the learned $Z_G$ to integrate both short-range and long-range dependencies between brain ROIs by introducing long-range embedding. This design allows our method to adaptively represent the communication connectivities in human brains.

**Readout Module.** To accomplish brain network analysis tasks, we take the output $Z_G$ of the self-attention module as the criterion, and then utilize an efficient readout function to derive the entire brain graph representation to further enhance the performance. In addition, we train an additional classifier for downstream tasks. The final classification basis is obtained as follows:

$$Y_G = \text{Softmax} \left( \text{MLP} \left( \text{Readout} \left( Z_G \right) \right) \right). \tag{10}$$

In Section 4.2, we evaluate the performance of various pooling methods. Ultimately, we employ clustering-based pooling as the readout function in the proposed method.

# 4 Experiments

In this section, we analyzed the following aspects to demonstrate the effectiveness of the proposed method and its capability to capture long-range dependencies within brain networks.

**Q1.** Does ALTER outperform other state-of-the-art models?

**Q2.** How does the proposed adaptive long-range aware strategy perform in different model architectures accompanied by various readout functions?

**Q3.** Does ALTER capture long-range dependencies within brain networks, and is ALGA strategy considered a key component?

## 4.1 Experimental Settings

**Datasets and Preprocessing.** We evaluate the proposed method using two brain network analysis-related fMRI datasets. 1) *Autism Brain Imaging Data Exchange* (ABIDE)[1], which contains 519 Autism spectrum disorder (ASD) samples and 493 normal controls. 2) *Alzheimer's Disease Neuroimaging Initiative* (ADNI)[2], which contains 54 Alzheimer's disease (AD) samples and 76 normal controls. During the construction of the brain graph, we first preprocess the fMRI data using the Data Processing Assistant for Resting-State Function (DPARSF) MRI toolkit. Next, we define brain ROIs based on predefined atlases from preprocessed fMRI data and calculate the average time-series feature for individual brain ROI. Finally, we formalize the brain graph $G = (V, X, A)$ for each sample according to the average time-series features of brain ROIs. Specifically, node features $X$ are functional connectivity matrix calculated by Pearson correlation, the adjacency matrix $A$ is the thresholded functional connectivity matrix to generate binary matrix of 0s or 1s, where the threshold is 0.3. The details of datasets and preprocessing can be found in Appendix A.

**Baselines.** The selected baselines correspond to two categories. The first category is generalized graph learning methods (Generalized - not specifies to brain networks), including SAN [30], Graphormer [31], GraphTrans [32], and LRGNN [42]. The second category (Specialized) is the brain graph-based methods , including BrainNetGNN [15], FBNETGEN [28], BrainGNN [12], BrainNETTF [15], A-GCL [43], and ContrastPool [44]. Note that the original code shared by the authors of these baselines is used for the comparative analysis. Please refer to the Appendix A for the details.

**Metrics.** Given the medical application of neurological disease classification tasks, we utilize both machine learning and medical diagnostic-specific metrics to evaluate the performance of the proposed method. These include classification Accuracy (ACC), Area Under the Receiver Operating Characteristic Curve (AUC), F1-Score, Sensitivity (SEN), and Specificity (SPE). In the experimental results, we report the mean and standard deviation across 10 random runs on the test dataset.

**Implementation Details.** In the proposed method, we set the number of steps $K$ for adaptive random walk to 16. The number of nonlinear mapping layers $L$ and attention heads $M$ of the self-attention module are set to 2 and 4, respectively. For all datasets, we randomly divide the training set, evaluation set and test set by the ratio of $7 : 1 : 2$. In the train processing, we adopt Adam as optimizer and CosLR as scheduler by a initial learning rate of $10^{-4}$ and a weight decay of $10^{-4}$. The batch size is set to 16 and the epoch is set to 200. All experiments are implemented using the PyTorch framework, and computations are performed on one Tesla V100.

## 4.2 Performance Comparison

In this sections, we evaluate the performance of ALTER by comparison with existing baselines to address **Q1**.

**Results.** Table 1 reports the comparison results between the proposed and the baseline methods. ALTER is able to significantly outperform the two categories of baseline methods for both datasets. In comparison to generalized graph learning methods, the proposed ALTER exhibits a significant improvement in terms of the ACC metric (10.9% improvement on the ABIDE dataset and 6.8% improvement on the ADNI dataset). For the case specialized graph learning methods, we again demonstrated superiority on both datasets in terms of the ACC metric (6.0% improvement on the ABIDE dataset and 5.1% improvement on the ADNI dataset). The reason for this performance improvement is that our method takes into account the communication strengths among brain ROIs

---

[1]http://preprocessed-connectomes-project.org/abide/
[2]https://adni.loni.usc.edu/

Table 1: Performance comparison with two categories of baselines on the two chosen datasets (%). The best results are marked in bold and the standard deviations are in parentheses.

| Category | Method | ABIDE | | | | ADNI | | | |
|---|---|---|---|---|---|---|---|---|---|
| | | AUC | ACC | SEN | SPE | AUC | ACC | SEN | SPE |
| Generalized | SAN | 71.3 (2.1) | 65.3 (2.9) | 55.4 (9.2) | 68.3 (7.5) | 68.1 (3.4) | 62.6 (5.2) | 52.4 (6.2) | 63.3 (8.5) |
| | Graphormer | 63.5 (3.7) | 60.8 (2.7) | **78.7 (22.3)** | 36.7 (23.5) | 60.6 (5.2) | 55.7 (3.1) | 60.1 (11.3) | 47.7 (13.5) |
| | GraphTrans | 60.1 (6.7) | 57.8 (4.7) | 65.7 (10.3) | 49.7 (11.5) | 61.2 (3.7) | 58.3 (5.1) | 66.2 (7.2) | 49.3 (3.1) |
| | LRGNN | 70.3 (4.1) | 66.1 (2.5) | 58.4 (9.2) | 65.2 (6.8) | 71.5 (6.4) | 67.3 (2.1) | 59.6 (1.2) | 49.7 (2.3) |
| Specialized | FBNETGEN | 75.6 (1.2) | 68.0 (1.4) | 64.7 (8.7) | 62.4 (9.2) | 73.5 (3.9) | 65.0 (2.6) | 61.3 (2.1) | 59.7 (1.2) |
| | BrainNetGNN | 55.3 (1.9) | 51.2 (5.4) | 67.7 (37.5) | 33.9 (34.2) | 53.7 (7.2) | 50.1 (2.1) | 64.2 (6.8) | 43.8 (8.0) |
| | BrainGNN | 71.6 (1.6) | 75.1 (3.2) | 69.4 (5.2) | 63.4 (7.1) | 63.5 (2.5) | 61.5 (3.2) | 65.1 (3.4) | 53.5 (4.1) |
| | BrainNETTF | 80.2 (1.0) | 71.0 (1.2) | 72.5 (5.2) | 69.3 (6.5) | 76.5 (2.4) | 69.0 (2.7) | 64.7 (7.1) | **75.0 (8.1)** |
| | A-GCL | 53.8 (0.5) | 53.8 (0.6) | 62.3 (5.0) | 54.5 (6.3) | 57.2 (1.1) | 52.2 (0.8) | 57.6 (42.4) | 52.6 (38.2) |
| | ContrastPool | 57.3 (0.8) | 57.4 (0.6) | 57.6 (6.8) | 57.0 (7.7) | 68.5 (3.2) | 69.2 (3.9) | 61.5 (17.2) | **75.4 (21.3)** |
| Ours | ALTER | **82.8 (1.1)** | **77.0 (1.0)** | 77.4 (3.4) | **76.6 (4.6)** | **78.8 (2.1)** | **74.1 (2.5)** | **76.5 (6.1)** | 70.0 (6.5) |

and utilizes this characteristic to guide the capture of long-range dependencies within the brain network.

**Vairous Readout Function.** In the experimental setups with and without the ALGA strategy, we compare the results of ALTER using various readout functions, including max pooling, sum pooling, average pooling, sort pooling [45], and clustering-based pooling [15]. As illustrated in Table 2 and Figure 3(a), our method employing the ALGA strategy consistently achieved superior performance across all readout function settings. Particularly, the combination of clustering-based pooling and the ALGA strategy yielded the best results. This phenomenon also addresses **Q2**.

## 4.3 Ablation Study

In order to assess the performance of the model from the **Q2** perspective, we conduct ablation studies on the ALGA strategy. This included performing evaluations with different architectures and readout functions.

**Adaptive Long-range Aware with Varying Architectures.** To verify the generalisability and effectiveness of the ALGA strategy, we implement it within different architectures, including SAN and Graphormer on the two selected datasets. As shown in Table 2, we find that the ALGA strategy can be adapted to different architectures, where this adaptation effectively improves the predictive power of models. This result demonstrates the effectiveness of this strategy in capturing long-range dependencies within brain networks. It should be noted that this analysis could only be performed on the generalized graph learning methods.

Table 2: Performance comparison with varying architectures on the two chosen datasets (%). The best results are indicated by underlining and the standard deviations are in parentheses.

| Method | ABIDE | | | | ADNI | | | |
|---|---|---|---|---|---|---|---|---|
| | AUC | ACC | SEN | SPE | AUC | ACC | SEN | SPE |
| Graphormer | 63.5 (3.7) | 60.8 (2.7) | 78.7 (22.3) | 36.7 (23.5) | 60.6 (5.2) | 55.7 (3.1) | 60.1 (11.3) | 47.7 (13.5) |
| Graphormer+ALGA | 67.2 (2.5) | 64.1 (1.9) | 82.3 (10.3) | 45.9 (12.7) | 62.9 (4.1) | 60.5 (2.9) | 63.5 (4.1) | 65.4 (2.9) |
| SAN | 71.3 (2.1) | 65.3 (2.9) | 55.4 (9.2) | 68.3 (7.5) | 68.1 (3.4) | 62.6 (5.2) | 52.4 (6.2) | 63.3 (8.5) |
| SAN +ALGA | 72.5 (1.9) | 67.8 (3.1) | 58.9 (6.5) | 70.8 (4.1) | 70.1 (2.3) | 65.8 (3.7) | 55.9 (4.8) | 68.3 (6.2) |
| ALTER w/o ALGA | 80.2 (1.0) | 71.0 (1.2) | 72.5 (5.2) | 69.3 (6.5) | 76.5 (2.4) | 69.0 (2.7) | 64.7 (7.1) | 75.0 (8.1) |
| ALTER | 82.8 (1.1) | 77.0 (1.0) | 77.4 (3.4) | 76.6 (4.6) | 78.8 (2.1) | 74.1 (2.5) | 76.5 (6.1) | 70.0 (6.5) |

**Adaptive Long-range Aware with Varying Readout Functions.** To further demonstrate that the ALGA strategy plays a key role in the proposed method, we take the ABIDE dataset as a benchmark and attempt to vary the readout function under varying architectures (only the AUC metric is shown, the full result can be referred to the Appendix A). Specifically, we first take the proposed method, SAN, and Graphormer as the basic framework, then employ five approaches including max pooling, sum pooling, average pooling, sort pooling, and clustering-based pooling as the readout functions. This analysis will reveal the prediction ability of the frameworks with varying readout functions in the presence and absence of ALGA strategy. The results of different readout functions under various

architectures are shown in Figure 3. We observe that, for any arbitrary readout function, the ALGA strategy enhances the performance of downstream tasks for various architectures compared to those without ALGA.

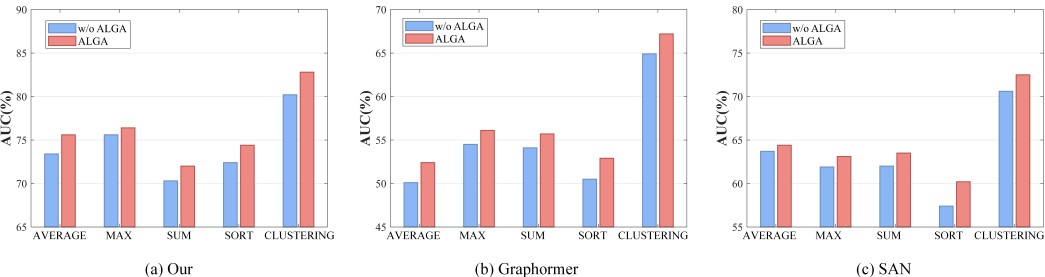

(a) Our                           (b) Graphormer                           (c) SAN

Figure 3: Performance comparison with varying readout functions (%).

## 4.4  In-depth Analysis of ALTER and ALGA Strategy

Here, we delve into the analysis of ALGA strategy and present cases to evaluate long-range dependencies to demonstrate the effectiveness of ALTER to assess **Q3**.

First, we investigate the impact of key hyperparameter of ALGA strategy of ALTER, which is the number of hops. In this experiment, we set the number of hops of ALGA to 2, 4, 8, 16, 32 for both selected datasets (only the AUC is shown here, the full result can be found in Appendix A). Figure 4(a) clearly shows that for both datasets, the predictive power of the proposed method generally increases as the number of hops increases. This phenomenon can be attributed to the presence of long-range connectivity in communication and information processing within human brains, which our model effectively captures. Ignoring this characteristic will adversely affect the predictive power of graph learning methods in brain network analysis tasks.

Next, we investigate the impact of the adaptive factors in the ALGA strategy. Specifically, we remove the adaptive factors and observe the change in the predictive ability of the proposed method. The experimental results demonstrate (Figure 4(a)) the performance of the proposed method is degraded when adaptive factors are not used to adjust the adaptive long-range encoding. The underlying reason for this result is that inter-ROI correlations play a crucial role in reflecting the communication strengths among brain ROIs. Treating pairwise ROI connectivity equally could potentially have a detrimental effect on brain network analysis tasks that depend on inter-ROI communication.

Finally, we present the cases to demonstrate ALTER's ability to capture long-range dependencies within brain networks. In this experiment, we randomly sample an example brain graph from the ABIDE test set and used it to train our model to learn the corresponding node features without pooling operation, and thus compute the attention scores among the node features. Figure 4(c) illustrates one example graph and the corresponding attention heatmap (Figure 4(b)). More sample-level and group-level examples can be found in Appendix A. The attention heatmap demonstrates the communication patterns necessary for brain network analysis tasks. Specifically, certain ROIs receive higher attention scores from multiple other ROIs, irrespective of the distance between them. In particular, ROI 6 and ROI 19 present higher attention score, despite the fact that these two ROIs are 5 hops apart (Figure 4(c)).

## 5  Discussions and Conclusion

**Limitations.**  On one hand, while utilizing the brain graph transformer to integrate both short-range and long-range dependencies among brain ROIs, we still cannot ensure an optimal balance between them. In future research, we will explore how to achieve a better balance between short-range and long-range dependencies in brain network analysis, with the aim of achieving better research results. On the other hand, the experimental data we currently employ is limited to fMRI data. Although utilizing these data can demonstrate the crucial role of long-range dependencies in brain network analysis tasks, other forms of data, such as DTI data, are also worthy of exploration. In future work, we will delve into alternative forms of data and propose corresponding methods for capturing long-range dependencies within brain networks.

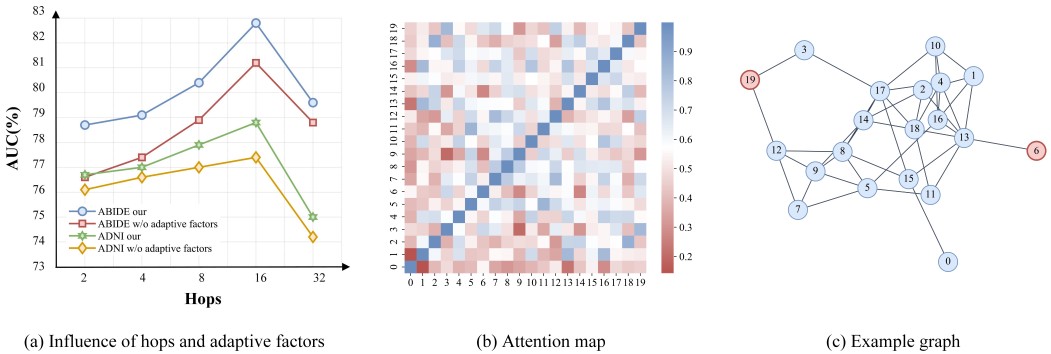

| (a) Influence of hops and adaptive factors | (b) Attention map | (c) Example graph |

Figure 4: In-depth analysis of ALTER and adaptive long-range aware strategy.

**Conclusion.** In summary, we present the ALTER model for brain network analysis, a novel brain graph transformer that explicitly captures long-range dependencies in brain networks and adaptively integrates them with short-range dependencies. Extensive experiments on ABIDE and ADNI datasets demonstrate that ALTER consistently outperforms generalized and specialized (specific to brain network analysis method) graph learning methods. This study presents an initial attempt to capture long-distance dependencies within brain networks and provides a new insight into understanding brain-wide communication and information processing.

## Acknowledgments and Disclosure of Funding

This work is supported by the National Natural Science Foundation of China (No.U21A20473, No. 62172370) and the Jinhua Science and Technology Plan (No. 2023-3-003a). Thanks to Guangqing Bai and Yuelong Huang for their help with baselines during rebuttal.

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

# A   Additional Experiments

Due to page limitations, only the experimental results for the AUC metric are included in the main text. For a more comprehensive study, the full results of the comparison experiments are provided in this appendix.

**Datasets.**   We preprocess the fMRI using the Data Processing Assistant for Resting-State Function (DPARSF) MRI toolkit. Specifically, we removed the first 10 time points from the downloaded nii data according to the default mode and chose slice timing, where the middle layer was the reference slice. meanwhile, we set the head motion correction to 'Friston 24', and selected automask and Nuisance covariates regression. the others were basically set according to the default mode. Then, considering individual differences, we choose to perform 'Normalize by DARTEL', and for the definition of ROIs, we adopt the altas already available in DPARSF. Finally, we construct brain networks for each fMRI. For parcellation, we utilize the Craddock 200 atlas, which defines 200 ROIs, for the ABIDE dataset. For the ADNI dataset, we apply the AAL atlas, comprising 90 cortical ROIs and 26 cerebellar ROIs.

**Baselines.**   The selected baselines correspond to two categories. The first category is generalized graph learning methods, including SAN [30], Graphormer [31], GraphTrans [32], and LRGNN [42]. SAN and Graphormer are two more popular Transformer-based graph learning methods. GraphTrans is capable of capturing long-range dependencies within general graphs. LRGNN combines neural architecture search to extract the long-range dependencies. The second category is the brain graph-based methods, including BrainNetGNN [15], FBNETGEN [28], BrainGNN [12], BrainNETTF [15], A-GCL [43], and ContrastPool [44]. BrainNetGNN utilized attention-based GNN to learn the representation of brain networks. FBNETGEN employed task-aware GNN, and BrainGNN adopted GNNs with ROI-aware and ROI-selection. BrainNETTF modeled a specific Transformer and readout function for brain network analysis. A-GCL utilizes adversarial graph contrastive learning to extract invariant features from brain networks. ContrastPool is capable of generating task-relevant, interpretable brain network representations. Although these methods show advantages in brain network analysis tasks, none of them analyzed and captured long-range dependencies within brain networks.

To ensure a fair comparison, we use the open-source codes of BrainGNN [12], BrainNETTF [15], FB-NETGEN [28], A-GCL [43], and ContrastPool [44]. For SAN [30], Graphormer [31], LRGNN [42], and GraphTrans [32], we adapt their open-source codes and modify them to suit the brain network datasets. For BrainNetGNN, we implemente it ourselves following the settings described in the paper. During the parameter tuning, we follow the tuning of BrainNETTF [15] for SAN, BrainGNN, FBNETGEN, Graphormer, and BrainNETTF. For BrainNetGNN, we search the number of GRU layers 1, 2, 3. For LRGNN, we vary the aggregation operations 8, 12 with the number of cell 1, 3. For GraphTrans, we search the number of GNN layers 1, 2, 3, 4 with the hidden dimension of 100. Regarding the construction of brain graphs for these baselines, we utilized functional connectivity matrix to compute a brain graph for BrainNETTF, which is computed by calculating the correlation between brain ROIs using the processed fMRI. The details of computing these correlation matrices is also incorporated to the revised paper. For BrainNetGNN and FBNETGEN, the models required the processed fMRI as input. ContrastPool, A-GCL, BrainGNN, SAN, Graphormer, LRGNN, and GraphTrans required the correlation matrix and adjacency matrix. As mentioned in the paper, the adjacency matrix is obtained by thresholding ($\geq 0.3$) the correlation matrix.

**Adaptive Long-range Aware with Varying Readout Functions.**   The full results of the ablation study on the three frameworks, VanillaTF, Graphormer, and SAN, are presented in Table 3, 4, and 5. Based on the results from the three tables, it is evident that, with the aid of the ALGA strategy, each framework employing various readout functions demonstrates superior performance across AUC, ACC, and SPE metrics. Furthermore, we observe that each framework based on clustering-based pooling exhibits significantly lower standard deviation in performance metrics when utilizing the ALGA strategy compared to those without it.

**The Impact of Hops and Adaptive Factors.**   In Table 6 and 7, we present the results of four metrics under different hops and with/without adaptive factors. Based on the results from the both tables, we observe that the proposed method typically achieves better performance on AUC and ACC metrics when using adaptive factors. Besides, we observe that, across both datasets, the proposed

Table 3: Performance comparison with varying readout functions on the VanillaTF framework (%). The overall best results are highlighted in bold, while better results with/without ALGA are indicated by underlining. Standard deviations are presented in parentheses.

| Readout | w/o ALGA | | | | ALGA | | | |
|---|---|---|---|---|---|---|---|---|
| | AUC | ACC | SEN | SPE | AUC | ACC | SEN | SPE |
| MEAN | 73.4(1.4) | 67.4(1.2) | 68.3(1.2) | 66.7(1.2) | 75.6(1.6) | 71.6(1.3) | 70.8(1.2) | 69.4(1.2) |
| MAX | 75.6(1.4) | 68.4(1.3) | 69.5(1.4) | 67.7(1.2) | 76.4(1.7) | 72.6(1.4) | 71.7(1.3) | 70.7(1.1) |
| SUM | 70.3(1.6) | 62.4(1.3) | 63.6(1.4) | 67.6(1.2) | 72.0(1.2) | 68.3(1.3) | 68.6(1.2) | 67.5(1.3) |
| SORT | 72.4(1.3) | 65.2(1.2) | 66.0(1.2) | 65.3(1.3) | 74.4(1.4) | 69.8(1.2) | 69.9(1.3) | 69.1(1.2) |
| CLUSTERING | 80.2(1.0) | 71.0(1.2) | 72.5(5.2) | 69.3(6.5) | **82.8(1.1)** | **77.0(1.0)** | **77.4(3.4)** | **76.6(4.6)** |

Table 4: Performance comparison with varying readout functions on the Graphormer framework (%). The overall best results are highlighted in bold, while better results with/without ALGA are indicated by underlining. Standard deviations are presented in parentheses.

| Readout | w/o ALGA | | | | ALGA | | | |
|---|---|---|---|---|---|---|---|---|
| | AUC | ACC | SEN | SPE | AUC | ACC | SEN | SPE |
| MEAN | 50.1(1.1) | 48.6(2.2) | 69.1(5.7) | 39.6(6.2) | 52.4(1.4) | 51.8(1.4) | 72.7(5.2) | 43.6(4.4) |
| MAX | 54.5(3.6) | 53.3(2.1) | 74.7(5.2) | 40.4(24.2) | 56.1(1.4) | 55.8(1.3) | 73.9(6.2) | 43.9(14.5) |
| SUM | 54.1(1.3) | 53.9(1.4) | 74.6(4.4) | 39.7(12.2) | 55.7(1.6) | 57.8(2.1) | 73.3(5.2) | **50.7(10.2)** |
| SORT | 50.5(4.7) | 49.6(5.2) | 76.5(6.4) | 40.6(19.3) | 52.9(5.3) | 53.9(5.4) | 80.7(11.3) | 44.6(9.3) |
| CLUSTERING | 64.9(2.7) | 60.3(3.3) | 79.4(12.5) | 41.7(20.1) | **67.2(2.5)** | **64.1(1.9)** | **82.3(10.3)** | 45.9(12.7) |

method generally performs exceptionally well with a hop count of 16. Hence, we set the number of hop to 16 in our comparison experiments.

**The Sensitivity of ALTER.** We present experimental results on the ABIDE dataset for hop counts $k$ ranging from 2 to 16 to analyze the sensitivity of ALTER. As shown in Table 8, we observe that as the number of hops increases, ALTER generally exhibits improved performance, achieving the best results at 16 hops. This indicates that our method is influenced by the number of hops $k$, as ALTER relies on random walk sampling to capture long-range dependencies. Additionally, this phenomenon demonstrates the capability of the proposed method to capture long-range dependencies.

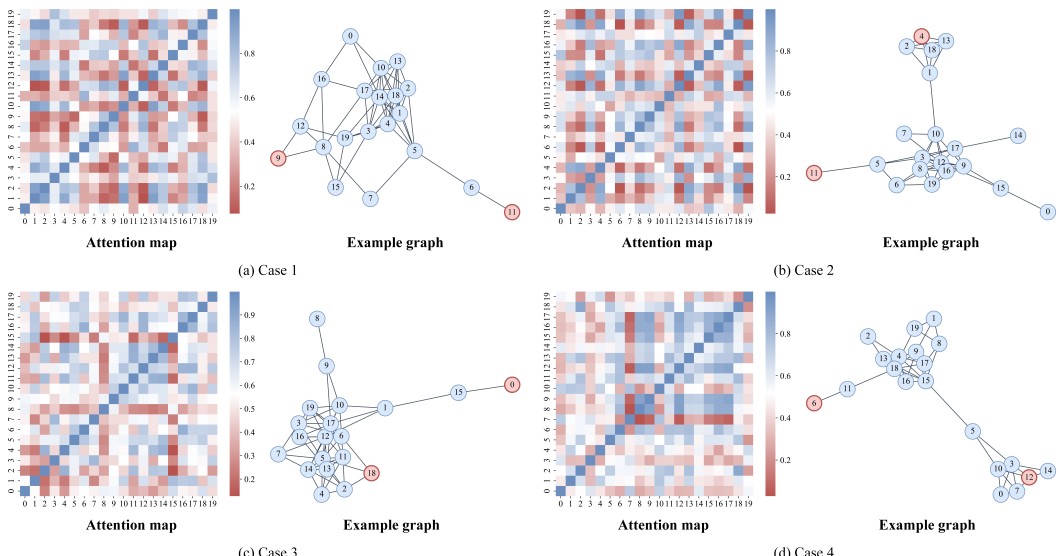

(a) Case 1      (b) Case 2

(c) Case 3      (d) Case 4

Figure 5: Example brain graphs from the ABIDE dataset and the corresponding attention heatmap.

Table 5: Performance comparison with varying readout functions on the SAN framework (%). The overall best results are highlighted in bold, while better results with/without ALGA are indicated by underlining. Standard deviations are presented in parentheses.

| Readout | w/o ALGA | | | | ALGA | | | |
|---|---|---|---|---|---|---|---|---|
| | AUC | ACC | SEN | SPE | AUC | ACC | SEN | SPE |
| MEAN | 63.7(2.4) | 60.7(3.3) | 56.7(1.3) | 58.6(2.4) | 64.4(1.6) | 62.8(2.5) | 57.6(1.5) | 63.4(3.6) |
| MAX | 61.9(2.5) | 56.9(2.9) | 54.2(3.1) | 52.4(4.2) | 63.1(1.7) | 59.4(1.2) | 54.1(3.4) | 59.9(3.1) |
| SUM | 62.0(2.3) | 57.8(2.6) | 55.7(3.2) | 60.7(4.7) | 63.5(1.2) | 60.7(1.4) | 57.8(2.4) | 61.6(5.2) |
| SORT | 57.4(5.2) | 55.6(5.2) | 53.7(4.3) | 56.7(3.2) | 60.2(1.4) | 58.9(4.7) | 56.5(4.5) | 59.8(3.6) |
| CLUSTERING | 70.6(2.4) | 67.3(3.4) | 56.7(7.5) | 67.6(12.4) | **72.5(1.9)** | **67.8(3.1)** | **58.9(6.5)** | **70.8(4.1)** |

Table 6: The Impact of Hops and Adaptive Factors on the ABIDE dataset (%). The overall best results are highlighted in bold, while better results with/without adaptive factors are indicated by underlining. Standard deviations are presented in parentheses.

| Hops | w/o adaptive factors | | | | adaptive factors | | | |
|---|---|---|---|---|---|---|---|---|
| | AUC | ACC | SEN | SPE | AUC | ACC | SEN | SPE |
| 2 | 76.6(3.9) | 69.0(2.5) | 70.1(3.5) | 69.2(5.1) | 78.7(3.9) | 72.0(2.0) | 71.9(3.1) | 72.2(3.1) |
| 4 | 77.4(3.1) | 71.0(3.0) | 72.5(2.5) | 71.1(4.5) | 79.1(2.4) | 73.0(2.5) | 73.2(1.9) | 72.9(3.8) |
| 8 | 78.9(2.2) | 73.0(2.0) | 74.2(2.9) | 75.4(5.9) | 80.4(1.9) | 72.0(4.0) | 75.6(2.1) | 74.8(5.6) |
| 16 | 81.2(2.6) | 75.0(1.5) | 75.8(3.1) | 75.1(4.2) | **82.8(1.1)** | **77.0(1.0)** | **77.4(3.4)** | **76.6(4.6)** |
| 32 | 78.8(1.9) | 74.0(1.2) | 76.4(2.2) | 73.8(5.9) | 79.6(2.1) | 75.0(1.3) | 76.3(2.7) | 74.3(5.1) |

**The Cases of the ALTER.** To further illustrate the ability of the proposed method to capture long-range dependencies within brain networks, we perform sample-level and group-level analyses, respectively. For the sample-level analysis, we present four additional cases in the Figure 5. From the Figure 5(d), we observe that despite being separated by 6 hops, ALTER is still able to capture the dependency between nodes 6 and 12. For the group-level analysis, we have computed the average across individuals to perform group-level analysis, as this approach aligns with the methodologies commonly adopted in similar studies [46]. The average graph and the corresponding attention heatmap are illustrated in Figure 6(b)&(c) of the global response. We can observe that ALTER captures group-level long-distance dependence, but it is not very significant relative to the individual-level. This may be due to certain individual differences in patients, including age and gender, which can affect the brain-wide communication [1].

**The Interpretability of ALTER.** We use the SHAP model for interpretability analysis on the ADNI dataset. We calculate the SHAP values of the attention matrix. From the Figure 6(a), it can be observed that the hippocampal regions of AD cases have positive SHAP values and the Top-10 ROIs with the highest SHAP values are almost always correlated with ADNI prediction, which is generally consistent with the results in [47].

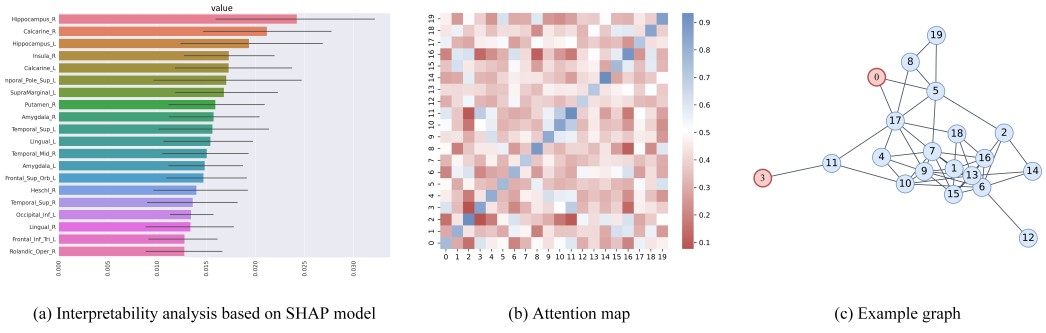

(a) Interpretability analysis based on SHAP model          (b) Attention map          (c) Example graph

Figure 6: The interpretability analysis and group-level analysis.

Table 7: The Impact of Hops and Adaptive Factors on the ADNI dataset (%). The overall best results are highlighted in bold, while better results with/without adaptive factors are indicated by underlining. Standard deviations are presented in parentheses.

| Hops | w/o adaptive factors | | | | adaptive factors | | | |
|---|---|---|---|---|---|---|---|---|
| | AUC | ACC | SEN | SPE | AUC | ACC | SEN | SPE |
| 2 | 76.5(3.2) | 71.5(2.5) | 73.9(5.3) | 69.1(5.8) | 77.1(2.9) | 71.8(1.8) | 71.2(6.2) | 67.7(6.6) |
| 4 | 76.9(3.4) | 72.1(2.9) | 74.0(5.2) | 68.4(6.1) | 77.3(3.2) | 73.3(2.3) | 74.6(6.9) | 68.9(7.3) |
| 8 | 77.6(2.5) | 72.5(3.1) | 75.6(6.5) | 68.9(6.8) | 78.2(2.8) | 73.6(2.7) | 75.4(7.2) | 69.2(5.8) |
| 16 | 78.1(2.9) | 73.0(2.1) | 75.4(5.9) | **71.2(6.2)** | **78.8 (2.1)** | **74.1 (2.5)** | **76.5 (6.1)** | 70.0 (6.5) |
| 32 | 76.2(2.9) | 73.0(1.2) | 74.0(4.5) | 70.7(5.5) | 77.9(2.6) | 73.5(2.6) | 74.9(5.2) | 70.3(6.1) |

Table 8: The sensitivity of ALTER on the ABIDE dataset. The best results are highlighted in bold, while standard deviations are presented in parentheses.

| Hops | ABIDE | | | |
|---|---|---|---|---|
| | AUC | ACC | SEN | SPE |
| 2 | 78.7(3.9) | 72.0(2.0) | 71.9(3.1) | 72.2(3.1) |
| 3 | 77.4(4.8) | 70.4(2.2) | 70.3(4.7) | 71.3(5.1) |
| 4 | 79.1(2.4) | 73.0(2.5) | 73.2(1.9) | 72.9(3.8) |
| 5 | 78.6(4.9) | 70.2(4.1) | 72.3(3.2) | 72.2(4.8) |
| 6 | 76.6(4.3) | 71.5(3.5) | 72.0(2.2) | 69.5(6.8) |
| 7 | 78.0(3.2) | 69.2(2.0) | 72.9(2.1) | 70.0(2.6) |
| 8 | 80.4(1.9) | 72.0(4.0) | 75.6(2.1) | 74.8(5.6) |
| 9 | 79.8(2.9) | 74.2(3.1) | 76.0(2.8) | 71.7(3.4) |
| 10 | 81.1(2.1) | 73.4(2.9) | 71.7(6.2) | 73.7(5.8) |
| 11 | 77.4(3.6) | 71.0(3.0) | 76.4(3.2) | 71.4(6.3) |
| 12 | 80.9(2.1) | 74.0(3.5) | 74.5(3.6) | 72.2(4.2) |
| 13 | 80.7(3.1) | 73.2(3.2) | 74.8(5.1) | 73.4(5.2) |
| 14 | 80.8(1.6) | 75.0(2.0) | 72.7(4.5) | 70.5(6.5) |
| 15 | 79.3(3.2) | 75.5(2.5) | 72.2(4.6) | 69.6(6.1) |
| 16 | **82.8(1.1)** | **77.0(1.0)** | **77.4(3.4)** | **76.6(4.6)** |

# B   Further Discussion

**Possible Negative Societal Impacts.**   Given that the research in this paper involves neurological disease diagnosis, it is essential to declare the potential negative societal impacts of this study, even though it is currently in the research phase and has not yet been applied in practice. Specifically, in the process of AI-assisted disease diagnosis, erroneous results are inevitable. Such errors can have severe consequences for patients and society. Therefore, in real-world medical diagnostic scenarios, the final decision should always rest with the physician's diagnosis.

