# OpenReview forum: "Long-range Brain Graph Transformer"
_NeurIPS.cc/2024/Conference — NeurIPS 2024 poster_

### Official Review · Reviewer_s7Az · 2024-06-21

**Soundness:** 2
**Presentation:** 3
**Contribution:** 2
**Rating:** 5
**Confidence:** 5

**Summary:**

This paper employs a random walk approach to capture long-range dependencies in the brain through a feature engineering scheme. The computed adaptive factors are incorporated into node features and used to train a Transformer model.

**Strengths:**

1. The idea of capturing long-range dependencies in the brain is reasonable.
2. The paper is well-written and easy to understand.
3. The experimental results are strong on the two datasets tested.

**Weaknesses:**

1. Some design choices in the proposed method are not well-motivated:
   - Using the designed adaptive factors (Eq. 1) instead of Pearson correlations.
   - Introducing the degree matrix in the random walk kernel.
2. The parameter \( k \) in Section 3.2 is not clearly defined; it is assumed that \( k = K \).
3. The descriptions of the dataset construction and experimental settings are unclear. Specifically:
   - Which atlas is used, and how many ROIs does it contain?
   - What exact value is used to threshold the connectivity to obtain the adjacency matrix?
   - Given that the number of hops is set up to 32, it is expected that the adjacency matrix is super sparse. However, such a high threshold is uncommon in brain network analysis.
4. The paper conducts experiments on only two datasets, which is insufficient. The ADNI dataset used contains only two classes with 130 subjects. It would be better to extend the experiments to include more datasets as referenced in [1], which includes six datasets and over 1.3k subjects for ADNI.
5. There is a lack of comparison with the latest related works. For example:
   - [2] introduces a graph augmentation using ALFF features.
   - [3] presents a global matching-based graph kernel that captures dynamic changes in evolving brain networks.
   - [4] utilizes clustering-based graph pooling for readout.
6. It would be valuable to incorporate the proposed ALGA method with GNN baselines such as GCN and GAT to evaluate whether it can improve their performance.
7. The paper lacks deeper analysis from a neuroscience perspective. It is important to provide insights from the model results, such as identifying specific ROIs related to the target disease and whether these findings align with medical literature.

References:
[1] Data-driven network neuroscience: On data collection and benchmark. NIPS 2023
[2] A-GCL: Adversarial graph contrastive learning for fMRI analysis to diagnose neurodevelopmental disorders. MIA 2023
[3] Effective Graph Kernels for Evolving Functional Brain Networks. WSDM 2023
[4] Contrastive Graph Pooling for Explainable Classification of Brain Networks. IEEE TMI 2024

**Questions:**

1. Which atlas is used in the dataset?
2. Which ROIs obtain high attention, and do these match domain observations for the diseases studied?

**Limitations:**

The limitations have been discussed in this paper.

---

> ### Author Rebuttal · Authors · 2024-08-07
>
> We thank you for your detailed feedback and your questions. We hope we have well addressed your concerns. If there are any other issues remaining, we are pleased to address them further.
>
> **W1.a**
> In fact, we employ Pearson correlations as adaptive factors in random walk, without any modifcation. Since the correlation between ROIs reflects their communication strength, which is crucial for comprehending the functional organization, dysfunctions, and information propagation. We take the correlation between ROIs as adaptive factors to influence the exploration mechanism of random walk, which induces ROIs with stronger connectivities to exhibit higher transfer probabilities of the next hop. This manner simulates real-world brain-wide communication.
>
> We recognize that this part of the presentation may not have been clearly articulated, leading to potential misunderstandings. Therefore, we have make the necessary revisions in the revised paper to enhance clarity.
>
> **W1.b**
> The introduction of degree matrix can help to obtain richer information about brain-wide communication and is very commonly used in network analytics [1]. Since the degree matrix provides the number of degrees for each ROI, its ability to reflect the active state of the ROI in communication is important in determining which ROIs play a key role in information propagation. Hence, a degree matrix determines the transfer probability of a node to its neighboring nodes and highly influences the behavior of walker [2]. We have now incorporated these details in the revised paper.
>
> **W2**
> We apologize for our mistake, we did mix up $k$ and $K$. As you say, what we use is $k = K$. $k$ is generally denoted as the number of hops in random walks. We apologize for not defining it explicitly, which has now been addressed in the revised paper.
>
> **W3&Q1**
> While nearly all of the experimental settings are presented in the source code, we acknowledge that our descriptions of these settings were indeed insufficient. We have now incorporated the details in Appendix A.
>
> Specific details requested by the reviewer includes:
> - For the ABIDE dataset, we use the Craddock 200 atlas with 200 ROIs. For the ADNI dataset, we use the AAL atlas with 90 cortical ROIs and 26 cerebellar ROIs.
> - The threshold is 0.3.
> - This is a mistake in the number of hops reported in the paper. The experiments were performed with 16 hops. We tried the hops commonly used for long-range dependency capture in network analysis by random walks through a grid search, and chose 16 hops. As shown in Figure 4(a) in the paper, larger hops do not lead to better results since brain graphs are relatively dense.
>
> **W4&W5**
> Upon receiving comments, we promptly review the referenced papers and their corresponding open-source codes. As shown in Tables 1&2&3 of the global response, we have added additional datasets (PPMI, Matai, TaoWu, and Neurocon) and baselines (A-GCL [3] and ContrastPool [4]).
>
> Regarding the complete ADNI dataset [5], we discovered from the GitHub repository provided by the authors that due to data protocol issues, it has not yet been made publicly available. Regarding [6], we find that the original papers did not provide corresponding code links, and also not released in github. We also contacted the authors, but as of the end of the rebuttal, we have not received responses. However, we need to state that [6] more focuses on learning time series of fMRI using a global and local matching-based graph kernels to construct dynamic brain networks, whereas ALTER is more concerned with capturing long-distance dependencies from static brain networks using random walk kernel, which are fundamentally different from each other.
>
> In the field of neuroscience, it is often challenging to obtain large amounts of data due to various constraints. And often use a smaller dataset (≤130 individuals), which are sufficient to support conclusions [7, 8, 9, 10]. We initially focused only on the ADNI and ABIDE because they are widely used for disease prediction. However, due to the significant challenges in preprocessing, we were unable to process the complete fMRI data from the ADNI dataset. While only two classes and 130 subjects were included, our method still shows excellent performance. This demonstrates the effectiveness of ALTER with limited data, suggesting that it may better adapt to other datasets with similarly limited quantities. This adaptability can potentially lead to more successful outcomes in other brain science-related medical tasks.
>
> **W6**
> As shown in the Table 5 of the global response we did additional experiments and achieved better performance. This is because ALGA can capture the long-range dependencies in the brain network.
>
> **W7&Q2**
> Based on the SHAP model, we have supplemented the experiment in neuroscience perspective, and concluded that ALTER is can recognize disease-related regions and give high attention. The results are illustrated in Figure 1(a) of the global response.
>
> **References**
>
> [1] Graph neural networks with learnable structural and positional representations. In ICLR 2022.
>
> [2] How to Count Triangles, without Seeing the Whole Graph. In KDD 2020.
>
> [3] A-GCL: Adversarial graph contrastive learning for fMRI analysis to diagnose neurodevelopmental disorders. MIA 2023.
>
> [4] Contrastive graph pooling for explainable classification of brain networks. TMI 2024.
>
> [5] Data-driven network neuroscience: On data collection and benchmark. In NeurIPS 2023.
>
> [6] Effective graph kernels for evolving functional brain networks. In WSDM 2023.
>
> [7] Spatio-Temporal Graph Hubness Propagation Model for Dynamic Brain Network Classification TMI 2024.
>
> [8] RH-BrainFS: Regional Heterogeneous Multimodal Brain Networks Fusion Strategy. In NeurIPS 2023.
>
> [9] Functional brain network reconfiguration during learning in a dynamic environment. Nature Communications 2020.
>
> [10] Increased global integration in the brain after psilocybin therapy for depression. Nature Medicine 2022.

---

> > ### Comment · Reviewer_s7Az · 2024-08-08
> > **Follow-up Questions**
> >
> > I am glad to find that the authors have addressed most of my concerns.
> >
> > However, I still have some follow-up questions:
> >
> > 1. Regarding the threshold used for sparsifying the connectivity to obtain the adjacency matrix, does 0.3 mean retaining edges with Pearson correlation larger than 0.3 or keeping the top 30% of edges? Additionally, does this thresholding drop all negative edges, or are absolute values used?
> >
> > 2. For the new experiment conducted, are you using the same settings as the original paper, or have you applied your own settings to them?

---

> > > ### Author Response · Authors · 2024-08-08
> > >
> > > Thank you for your feedback. We would like to reply to your questions and comments as follow:
> > >
> > > **Q1.** We keep edges with Pearson correlation greater than 0.3. This threshold also removes the negative connections. This strategy aligns with the methodologies commonly adopted in previous studies [1, 2].
> > >
> > > **Q2.** Yes, we followed the settings in the original papers. Specifically, for ContrastPool, although it utilizes the same datasets as ours (PPMI, Matai, TaoWu, and Neurocon), we ensured fairness by using the thresholded correlation matrix as the adjacency matrix. In A-GCL, since we use different datasets than those in the paper, we performed a parameter search based on the paper's recommendations to ensure the fairness of the results. Where the batch size was searched from {8, 16, 32, 64}, the learning rate of the parameter µ was searched from {0.0001, 0.0005, 0.001, 0.005, 0.01}, and the learning rate of the parameter z was searched from {0.0005, 0.001 , 0.01}.
> > >
> > > We sincerely thank you for your efforts in reviewing our paper. We hope we have resolved all the concerns, and we will deeply appreciate it if you could reconsider the score accordingly. We are always willing to address any of your further concerns.
> > >
> > > [1] RH-BrainFS: Regional Heterogeneous Multimodal Brain Networks Fusion Strategy. In NeurIPS 2023.
> > >
> > > [2] BrainGNN: Interpretable Brain Graph Neural Network for fMRI Analysis. MIA 2021.

---

> > > > ### Comment · Reviewer_s7Az · 2024-08-08
> > > >
> > > > Thank you for all your efforts to address my concerns. I greatly appreciate them. I lean to accept this work now and have raised my rating. Please make sure all the discussed points are included in your revision.

---

> > > > > ### Author Response · Authors · 2024-08-08
> > > > >
> > > > > We really appreciate your valuable comments and positive recommendation with the new scores. All the discussed points will be explicitly clarified in future versions of the paper.

---

### Official Review · Reviewer_xvKE · 2024-07-07

**Soundness:** 3
**Presentation:** 4
**Contribution:** 3
**Rating:** 7
**Confidence:** 5

**Summary:**

This paper highlights a significant gap in the existing literature on brain network representation learning, specifically the inadequacy of current methods to effectively capture long-range dependencies, leading to limited an integrated understanding of brain-wide communication. To bridge this gap, the paper introduces ALTER, an innovative model designed for adaptive awareness of long-range dependencies and brain network representation learning. The model consists of a long-range aware strategy for capturing long-range dependencies and a transformer framework for integrating multi-level communication understanding.

**Strengths:**

(1) This paper is generally clear and well-written, providing a comprehensive analysis of the problem to be solved, supported by some convincing references, and emphasizing the urgent need to come up with solutions that can capture long-distance dependencies between brain ROIs. At the same time, the authors make the problem better understood with illustrations.
(2) The logic of the paper is smooth. The authors clarify the issue, after which they elaborate on the theoretical causes of the issue, and finally design a specific biased random walk strategy for the theoretical causes and obtain encouraging experimental results.
(3) I found the method presented here is technically sound with excellent results. From the experimental results, it is shown that the proposed adaptive long-range aware strategy is very effective in long-range dependencies capture and can greatly enhance the effectiveness of the disease diagnosis task. In addition, the ablation study conducted on various components reveals the superiority of the adaptive long-range aware strategy.
(4) The authors clearly and frankly understand the drawbacks and strengths of the method, e.g. the method does not allow for a good trade-off between long- and short-range. This makes this method a clear one to build on top of, prompting further interesting work in highly important domain.

**Weaknesses:**

1. Although the results of the ablation experiments verify that biased random walk is efficient, it is difficult to convince me of the necessity of introducing this module based only on the theoretical description of biased random walk in the methods section. As far as I know, random walk is common in graph representation learning. Therefore, I would like to ask the authors to provide me with a more in-depth analysis of biased random walk, otherwise I do not understand why biased random walk is effective in capturing long-range dependencies in brain networks.
2. Transformer-based graph representation learning is not original in brain network analysis. It is recommended that the authors provide a specific explanation for fusing long-range and short-range dependent embeddings using the graph Transformer.
3. Lack of interpretable analysis. Interpretive analysis of brain network representation learning models geared toward brain disease diagnosis is important. In the context of brain network analysis, I suggest that papers should explain and identify the brain regions or networks that are most relevant to the task of brain disease classification.

**Questions:**

1. Have the authors considered tasks other than binary categorization?
2. Am I correctly understanding that the dimension of the original feature embedding and the long-range dependency embedding after splicing is equal to the original feature embedding dimension? In the representation in Figure 2, I get the impression that the dimension after splicing is unchanged, but this does not seem to be the case from the text. Could the authors clarify this point?
3. The authors conducted experiments on fMRI data sets, and fMRI is only single-modal. Is the biased random walk strategy of ALTER still valid on multi-modal data?

**Limitations:**

See weaknesses and questions for details.

---

> ### Author Rebuttal · Authors · 2024-08-07
>
> We thank you for acknowledging the novelty of the proposed method and for suggesting relevant analysis, which we have included in the global response. We hope to provide satisfying answers to the concerns raised. If there are any other issues remaining, we are pleased to address them further.
>
> **W1**
> Our model aims to capture long-range dependencies within brain networks leveraging the biased random walk, which has a Markovian nature that captures long-range communication among brain ROIs by sampling and encoding sequences into embeddings. We agree that random walk is common in graph representation learning. However, thedifferent pairs of ROIs in brain networks usually exhibit different communication strengths in brain activity. As a result, traditional random walk methods are usually not applicable to brain networks, and cannot capture long-range dependencies in brain networks.
>
> **W2**
> We acknowledge that Transformer is commonly used in brain network analysis. The focus of the proposed method is on capturing long-range dependencies within brain networks. Transformer inherently has limitations in capturing long-range dependencies, which prevents a comprehensive understanding of brain-wide communication. Instead, the proposed ALTER method introduces an adaptive long-range aware strategy to explicitly capture long-range dependencies within brain networks. It then integrates long-range and short-range dependencies using the adaptive mechanism of Transformer, thereby achieving a multi-layered understanding of brain networks.
>
> **W3**
> As shown in Figure 1(a) of the global response we supplemented the interpretability analysis.
>
> **Q1**
> In the submitted manuscript, we evaluated performance solely on binary classification tasks using the ADNI and ABIDE datasets. To further demonstrate the effectiveness of the proposed method, we have conducted additional experiments on the PPMI dataset with four class, as suggested by reviewer s7Az. The experimental results are presented in the Table 1 of the global response. The results show that the proposed ALTER method achieved the best performance, as it effectively captures long-distance dependencies between brain ROIs.
>
> **Q2**
> Thank you very much for your question. Since we concatenate the long-range dependency embeddings directly with the original feature embeddings without any other operation, the dimensionality of the concatenated features is not equal to that of the original feature embeddings. We have now revised the paper to improve the clarity of the methodology.
>
> **Q3**
> Thank you very much for your question. The value of the biased random walk strategy in ALTER for multi-modal data is indeed a topic worth exploring. However, due to some fundamental differences between  functional and structural brain networks constructed based on fMRI and DTI, directly applying the biased random walk strategy to capture long-range dependencies in them may not be feasible and would require tailoring the ALTER model. Nonetheless, the biased random walk strategy can be adjusted by learning consistent and complementary communication patterns between functional and structural brain networks, which may enhance its adaptability to multimodal brain network data. We have now incorporated this as a future work of the study in the revised paper.

---

> > ### Comment · Reviewer_xvKE · 2024-08-10
> > **Reply to rebuttal**
> >
> > I appreciate the detailed clarifications and additional interpretable experiments provided. These have effectively addressed my previous concerns and underscored the significance of this work. Of course, I am inclined to increase my score and recommend acceptance of the paper.
> >
> > Overall, the manuscript is well written and represents a valuable contribution to the research field. I look forward to seeing the integration of the aforementioned discussion in the next version, particularly the interpretability analysis.

---

### Official Review · Reviewer_u6uE · 2024-07-11

**Soundness:** 3
**Presentation:** 3
**Contribution:** 3
**Rating:** 7
**Confidence:** 4

**Summary:**

The study employs the adaptive long-range aware graph transformer (ALTER) to tackle the challenge of weak comprehension in whole-brain communication, which arises from the failure to capture long-range dependencies in brain networks. Initially, the study encodes long-range dependencies into long-range embeddings through biased random walk sampling, thereby enriching the embeddings with information on long-range dependencies. Subsequently, the study takes into account the significance of both short- and long-range dependencies in brain network analysis tasks. It introduces the graph transformer, which uses a self-attention mechanism to integrate these dependencies between brain ROIs. The objective is to capture varying levels of communication connections within the human brain. Experimental results demonstrate that ALTER outperforms current SOTA graph learning baselines, achieving superior AUC and ACC scores on the ABIDE and ADNI datasets.

**Strengths:**

(1) This work is significant. How to facilitate the understanding of communication and information processing among brain ROIs is a key issue. The authors provide a clear overview of the necessity of long-range dependencies for understanding communication and information processing among brain ROIs, and propose an effective method to capture long-range dependencies in brain networks.

(2) The paper is aesthetically pleasing in its writing form, especially the diagrams and charts that make it easy for the reader to understand the exact process of the work done. Meanwhile, the authors provide detailed preprocessing and implementation details. As a result, this paper is highly reproducible.

(3) The paper provides complete experimental results. Besides comparisons with the baseline and ablation studies on modules, the appendix section validates the state-of-the-art of the adaptive long-range dependency aware strategy on multiple readout functions, further enhances reproducibility, and facilitates a deeper understanding of the method.

(4) ALTER is an ingeniously crafted framework that is able to adaptively perceive both long- and short-range dependencies in brain networks. In addition, ALTER interestingly takes inter-ROI correlations into account in the capture of long-range dependencies in brain networks.

**Weaknesses:**

I have identified three main weaknesses that need to be clarified during the rebuttal process, which is the reason I gave a weak acceptance despite the strengths. If my points are properly addressed, I will be happy to review my scores based on the results of the rebuttal process. I numbered the comments to facilitate discussion.

1. The technical contributions are neutral. The proposed ALTER seems to be a combination of the random walk and the graph transformer, but its novelty and difficulty are unclear.

2. ALTER obtained SPE scores that were 5% lower than BrainNETTF on the ADNI dataset and SEN scores that were 1.3% lower than Graphormer on the ABIDE dataset. This performance lag is significant, but the authors do not seem to have clearly explained the reasons for the SPE and SEN lags.

3. The authors state in line 202 that ALERT uses a linear layer to inject long-range dependencies into the brain graph transformer, but that direct collocation also accomplishes the above, and the authors do not state the necessity of introducing a linear layer.

**Questions:**

In addition to what I wrote in the "Weaknesses" section, I have a couple of questions:

1. How does the author define long-range dependence and short-range dependence?

2. Did the authors attempt to choose other correlation measures than the Pearson correlation coefficient to define the adaptive factors?

3. Could you elaborate on the need to introduce the graph transformer in the paper? Would replacing the graph transformer with a GNN affect the results?

4. Could the authors elaborate on the correlation between the attention map and the example graph in Figure 5?

**Limitations:**

The paper emphasizes the possible negative social impact of work. However, in my opinion, the negative societal impact is not only medical errors due to prediction errors, but should also include the negative ethical impact of the model. Adding ethical concerns would help the reader better understand the potential impact of the proposed methodology.

---

> ### Author Rebuttal · Authors · 2024-08-07
>
> We thank you for your detailed assessment of our work and for highlighting the merits of our approach, as well as the importance of the problem. We address all concerns below, if there are any other issues remaining, we are pleased to address them further:
>
> **W1**
> Brain networks are inherently dense, which has led to the increasing use of transformers for their analysis. However, a significant limitation of transformers is their inability to effectively capture long-range dependencies, which limits the integrated understanding of brain-wide communication. The proposed ALTER is concerned with the capture of long-range dependencies between ROIs, not the combination of random walk and graph transformer. In particular, we design the ALGA strategy, which simulates real-world brain-wide communication by utilizing adaptive factors to evaluate the communication strength between ROIs and capturing long-range dependencies in brain networks through a Markov process. We have now re-emphasized the novelty and difficulty of the proposed ALTER in the revised manuscript.
>
> **W2**
> We acknowledge that the proposed method exhibits a lower SPE on the ADNI dataset compared to BrainNETTF and a lower SEN on the ABIDE dataset compared to Graphormer. However, considering the standard deviation, our method demonstrates more stability, which can be attributed to its ability to capture long-range dependencies within the brain network. Moreover, our method significantly outperforms the baselines on other metrics across both datasets. In particular, on the ADNI dataset, our method shows a 10.3% improvement in the SEN metric compared to the sub-optimal method. We have provided a detailed explanation of this aspect in the revised manuscript.
>
> **W3**
> Thank you for requesting clarity on the linear layer. The main objective of introducing a linear layer to map the initial long-range embedding to the transformer is to enable end-to-end learning of the long-range embedding to participate in updates and enhance its expressiveness. This method is able to enhance the expressiveness of the random walk embedding by capturing multiple types of graph propagation, which in turn facilitates the integration of global information in the brain graph. We have now added these details in the revised paper. If we do not use a linear layer and directly concatenate the initial long-distance embedding with the original features, this reduces the expressiveness of the long-distance dependencies in the brain graph learning. We additionally performed ablation experiments on ABIDE dataset to demonstrate the effectiveness of introducing a linear layer.
>
> |Method |AUC|ACC|SEN|SPE|
> |--- |-----------| ----------- | -----------| ----------- |
> | w/o Linear| 81.0(1.5) | 76.4(2.3)| 73.6(5.8) | 74.4(5.4) |
> | ALTER |82.8 (1.1) | 77.0 (1.0) | 77.4 (3.4) | 76.6 (4.6) |
>
> **Q1**
> we utilize the proposed ALTER method to qualify these dependencies as they manifest in the data. In brain network analysis, long-range dependence and short-range dependence refer to interactions between neurons in the spatial dimension. Among them, short-range dependencies usually refer to interactions that occur in the same brain region or between anatomically neighboring brain regions. In contrast, long-distance dependence usually spans larger spatial brain regions and involves the information transmission between different functional regions. Such long-range dependencies assume the integration of global information, which plays a key role in complex cognitive functions. In the manuscript, we only deal with the capture of long-range dependence without defining it explicitly at the neuroscience level.
>
> **Q2**
> Indeed, we have not yet explored other correlation measures besides the Pearson correlation coefficient to define the adaptive factors in our study. The proposed method represents an initial attempt to capture long-range dependencies in brain networks. In future work, we plan to investigate some non-linear correlation measures, to further refine and potentially improve the adaptability of our method. It is worth noting that studies in neuroscience predominantly rely on Pearson correlation.
>
> **Q3**
> In ALTER, the self-transformer mechanism is introduced to adaptively integrate the long-range and short-range dependencies between ROIs. Meanwhile, at the suggestion of reviewer s7Az, we try to replace the transformer with 1-layer GCN and 2-layer GCN. The experimental results are shown in the Table 5, which it is proved that transformer usually achieves better results due to its ability of adaptive learning.
>
> **Q4**
> In Figure 5 we show additional examples of long-range capture at the individual level on the ABIDE dataset. In Figure 5(a)(b)(c) we can see that despite the fact that the two red labeled nodes are 5 hops apart, there is still a high attention value between the two ROIs. In Figure 5(d) we can see that there is a high attention value between 6 and 12 which are 6 hops apart.

---

> > ### Comment · Reviewer_u6uE · 2024-08-12
> >
> > Thanks for the response.  The reply has addressed my concerns.  Overall, this work is of high quality and offers a novel perspective compared to recent studies.  I support the acceptance of this work.

---

### Official Review · Reviewer_dPgM · 2024-07-17

**Soundness:** 3
**Presentation:** 3
**Contribution:** 3
**Rating:** 5
**Confidence:** 4

**Summary:**

This work proposes Adaptive Long-range aware TransformER (ALTER), a brain graph transformer to capture long-range dependencies between brain ROIs utilizing biased random walk.

**Strengths:**

1. This work introduces a novel brain graph transformer with adaptive long-range awareness, which leverages the communication strengths between ROIs to guide the capturing of long-range dependencies.
2. The result demonstrates that ALTER consistently outperforms generalized graph learning methods and other graph learning-based brain network analysis methods.

**Weaknesses:**

1. Even though the proposed ALTER is better than all selected baselines, some baselines like BrainGNN can provide both the prediction and interpretation. It is unclear whether the ALTER can also explain the important disease-specific pattern and find the biomarkers.
2. It is unclear how to implement the comparable baselines and how to build the brain graph for them.
3. It is unclear how to preprocess the fMRI in detail and how to conduct the quality control.

**Questions:**

1. What is the final loss function for this task?
2. How many times do you repeat the experiment? Have you conducted k-fold cross-validation to examine the result?
3. How about the ALTER’s ability to capture long-range dependencies in the patient's group instead of choosing one example to show in Figure 4? Can it get the same conclusion in the group-level analysis?

---

> ### Author Rebuttal · Authors · 2024-08-07
>
> We thank you for the comments on our work. Below we address the questions raised. We hope we have well addressed your concerns. If there are any other issues remaining, we are pleased to address them further.
>
> **W1**
> As shown in Figure 1(a) of the global response, we have analyzed the output using the SHAP model and confirmed that our results reflect AN-related ROIs. The proposed ALTER promotes an integrated understanding of brain-wide communication by capturing long-range dependencies and achieves superior performance in disease prediction tasks. Long-range dependencies, as complementary to short-range dependencies, can deliver unique insights into the organization and behavior of brain networks associated with neuropsychiatric disorders [1, 2, 3]. ALTER effectively captures long-range dependencies using the Markov process-based ALGA strategy, which has the ability to further explain the important disease-specific pattern and find the biomarkers. It is worth noting that the proposed model employs Transformers which are very well known for extracting the important features using its attention mechanism, which is effectively used for interpretability.
>
> **W2**
> We apologize for not providing a clear description of how to implement the comparable baselines and build the brain graphs for them. We have added a section in the appendix to clarify the implementation details of the comparable baselines and the construction of the brain graphs.
>
> Specifically, to ensure a fair comparison, we use the open-source codes of BrainGNN, BrainNETTF, and FBNETGEN. For SAN, Graphormer, LRGNN, and GraphTrans, we adapt their open-source codes and modify them to suit the brain network datasets. For BrainNetGNN, we implemente it ourselves following the settings described in the paper. During the parameter tuning, we follow the tuning of BrainNETTF [4] for SAN, BrainGNN, FBNETGEN, Graphormer, and BrainNETTF. For BrainNetGNN, we search the number of GRU layers {1, 2, 3}. For LRGNN, we vary the aggregation operations {8, 12} with the number of cell  {1, 3}. For GraphTrans, we search the number of GNN layers {1, 2, 3, 4} with the hidden dimension of 100.
>
> We utilized functional connectivity matrix to compute a brain graph for BrainNETTF, which is computed by calculating the correlation between brain regions using the processed fMRI. The details of computing these correlation matrices is also incorporated to the revised paper. For BrainNetGNN and FBNETGEN, the models required the processed fMRI as input. BrainGNN, SAN, Graphormer, LRGNN, and GraphTrans required the correlation matrix and adjacency matrix. As mentioned in the paper, the adjacency matrix is obtained by thresholding (≥0.3) the correlation matrix.
>
> **W3**
> We thank the reviewer for requesting a clear explanation of the preprocessing steps for fMRI. We preprocess the fMRI using the Data Processing Assistant for Resting-State Function (DPARSF) MRI toolkit. Specifically, we removed the first 10 time points from the downloaded nii data according to the default mode and chose slice timing, where the middle layer was the reference slice. meanwhile, we set the head motion correction to ‘Friston 24’, and selected automask and Nuisance covariates regression. the others were basically set according to the default mode. Then, considering individual differences, we choose to perform ‘Normalize by DARTEL’, and for the definition of ROIs, we adopt the altas already available in DPARSF. Then, we construct brain networks $G = \left( {V,X,A} \right)$ for each fMRI.
>
> During the experiment, for the ABIDE dataset, since we directly adopted the processed brain network from [4] and used it as the correlation matrix, its quality control refers to [4]. For the ADNI dataset, besides the above preprocessing, we performed head motion correction, slice timing correction, realigning and normalize. Please note that we have followed the standard protocol used by the other research studies to ensure that any bias and noise have been removed from the dataset.
>
> **Q1**
> The final loss function for this task is the cross-entropy loss as the model address a classification task. This is already implemented in our open-source code as shown. We have now revised the paper to improve the clarity of the methodology.
>
> **Q2**
> For all the experiments, we repeated them 10 times. Instead of k-fold cross-validation, we conducted repeated random split validation. To ensure that our results are trustworthy, we additionally performed 5-fold cross-validation (as shown in Table 4 in the global response).
>
> **Q3**
> We have computed the average across individuals to perform group-level analysis, as this approach aligns with the methodologies commonly adopted in similar studies [5]. The average graph and the corresponding attention heatmap are illustrated in Figure 1(b)&(c) of the global response. We can observe that ALTER captures group-level long-distance dependence, but it is not very significant relative to the individual-level. This may be due to certain individual differences in patients, including age and gender, which can affect the brain-wide communication [6].
>
> **References**
>
> [1] Space-independent community and hub structure of functional brain networks. NeuroImage 2020.
>
> [2] Long-range connections are more severely damaged and relevant for cognition in multiple sclerosis. Brain 2019.
>
> [3] Engineering brain assembloids to interrogate human neural circuits. Nature Protocols 2022.
>
> [4] Brain network transformer. In NeurIPS 2022.
>
> [5] Structure-function coupling in the human connectome: A machine learning approach. NeuroImage 2021.
>
> [6] Local structure-function relationships in human brain networks across the lifespan. Nature 2022.

---

### Author Rebuttal · Authors · 2024-08-07

We thank all the reviewers for their insightful assessment of our work, as well as the useful feedback and actionable suggestions they provided. We are pleased that they found our work to be meaningful (reviewer u6uE) and reasonable (reviewer s7Az), that the experimental results are strong (reviewers s7Az, dPgM and xvKE), and that the manuscript is clear (reviewer xvKE). We will incorporate their suggestions to improve the presentation in future revisions of the paper.

Below we provide additional empirical analysis based on the recommendations raised by the reviewers. Each reviewer's individual questions will be answered in separate responses. The new results include:

**Interpretability Analysis.** We used the SHAP model for interpretability analysis on the ADNI dataset. We calculated the SHAP values of the attention matrix. From the results, it can be observed that the hippocampal regions of AD cases have positive SHAP values and the Top-10 ROIs with the highest SHAP values are almost always correlated with ADNI prediction, which is generally consistent with the results in [1].

**More Sufficient Experiments.** we evaluate the proposed ALTER on additional datasets (PPMI, Matai, TaoWu, and Neurocon). Meanwhile, we add more models including A-GCL [2] and ContrastPool [3].

**More Ablation Studies.** (1) We used a 5-fold cross validation replacing the repeated random split validation for evaluating the proposed ALTER. (2) We evaluate the performance of the ALGA strategy in combination with different GNN baselines, including 1-layer GCN, 2-layer GCN, 1-layer GAT, and 2-layer GAT.

**Group-level Analysis.** We have computed the average across individuals to perform group-level analysis, and then presented the average graph and the corresponding attention heatmap.

**References**

[1] Multimodal deep learning for Alzheimer’s disease dementia assessment. Nature communications 2022.

[2] A-GCL: Adversarial graph contrastive learning for fMRI analysis to diagnose neurodevelopmental disorders. MIA 2023.

[3] Contrastive graph pooling for explainable classification of brain networks. TMI 2024.

---

### Decision · Program_Chairs · 2024-09-25

**Decision:**

Accept (poster)

**Comment:**

The paper introduces a novel brain graph transformer, ALTER, with adaptive long-range awareness, which effectively captures long-range dependencies among brain regions of interest (ROIs) by leveraging their communication strengths. The approach is well-motivated, addressing the critical issue of understanding communication and information processing among brain ROIs, and shows promising results, consistently outperforming other graph learning methods in brain network analysis. The method is technically sound, and the experimental results demonstrate its effectiveness across multiple datasets, with detailed ablation studies further validating the adaptive long-range dependency-aware strategy. The paper is well-written, with clear diagrams and charts that enhance understanding and reproducibility.

However, the paper has some limitations. ALTER's interpretability in identifying disease-specific patterns remains unclear compared to methods like BrainGNN. The technical novelty, involving random walks and graph transformers, could be better explained, with some design choices appearing under-motivated. Details on baseline implementation, fMRI preprocessing, and brain graph construction are lacking, which could affect reproducibility. Additionally, while ALTER shows good performance, its scalability and comparative advantage over all baselines require further evaluation.

Despite these weaknesses, the strengths of the paper, particularly in addressing a key problem with a well-supported and innovative approach, justify its acceptance.